# Intraoperative Rupture of an Intracranial, Extradural Hydatid Cyst: Case Report and Treatment Options

**DOI:** 10.3390/brainsci11121604

**Published:** 2021-12-02

**Authors:** Cosmin-Nicodim Cindea, Vicentiu Saceleanu, Adriana Saceleanu

**Affiliations:** 1Faculty of Medicine, Lucian Blaga University of Sibiu, 550024 Sibiu, Romania; cindea.cos@gmail.com (C.-N.C.); Adriana.saceleanu@ulbsibiu.ro (A.S.); 2Department of Neurosurgery, County Clinical Emergency Hospital of Sibiu, 550245 Sibiu, Romania

**Keywords:** hydatid cyst, intracranial extradural, rupture, case report, craniotomy

## Abstract

A 23-year-old woman was presented to the Emergency Unit with intracranial hypertension syndrome and blindness in her left eye which had started recently. A cranial native computed tomography scan and a magnetic resonance imaging (MRI) with contrast examinations revealed a giant intracranial cystic lesion, extending into the left frontal lobe, which was compressing the optic chiasm and eroding the internal plate of the left frontal bone. Surgical craniotomy was performed for evacuation and decompression, but during the craniotomy the cyst ruptured. After assessing the degree of erosion of the internal bone plate, we concluded that the primary origin of the cyst was intraosseous. With the dura mater being intact, abundant lavage with H_2_O_2_ was applied and the bone flap was replaced after rigorous bone scraping. Imaging control at six and twelve months identified no recurrence of the cyst. In the literature, hydatid cysts located in the skull bone are very rare and most of them rupture intraoperatively. Given their extremely low incidence in developed countries, any neurosurgeons’ experience with such pathology is limited and in some cases surgery cannot be delayed. In the case of intracerebral hydatid cysts, a neurosurgeon usually has only one shot at surgery, so simple and quick-to-access therapeutic guidelines must be developed in order to inform the choice of surgical technique. We conclude that the most successful surgical approach could be double concentric craniotomy. This surgical technique is used in intracerebral tumors, which also have an important bone invasion.

## 1. Introduction

Cystic Echinococcosis, known as hydatidosis or hydatid disease, is cause by infection with Echinococcus granulosus [1].

Echinococcosis is a zoonosis transmitted from animals to humans mainly through a pastoral cycle [2]. Intermediate hosts are most often sheep, cattle or pigs. Contaminated viscera are ingested by the final hosts, most often dogs, who will develop tapeworm echinococcus in their digestive system. Tapeworm eggs are excreted in the feces and can be accidentally ingested by humans through contaminated food, insufficiently prepared food or dirty hands. After the infection occurs, one or more cysts will develop, most often in the liver or lungs (70–90%), but other locations are possible (10–30%), including the central nervous system, bones, kidneys, muscle and spleen.

The number of new cases in the European Union is about 800 cases/year, but it is expected that a large number do not even reach the official statistics. The result is the underestimation of the magnitude of this pathology [3]. This understating is also due to the slow evolution of the pathology. Although infection can occur in childhood, it can become symptomatic later on, and combined with the fact that the rural population in isolated areas is especially affected, we can expect to lose many cases in the statistics [4].

We know from other studies that the countries located in the south and east of the European Union have the highest incidence. In the east of the European Union, the incidence has decreased due to better diagnoses. There were also important mass screening projects, such as those in the WHO-IWGE (WHO Informal Working Group on Echinococcosis) and HERACLES (The Human Cystic Echinococcosis ReseArch in Central and Eastern Society. The WHO-IWGE study performed an ultrasound screening on a group of 4146 subjects from rural areas in Romania, identifying 173 patients with hydatid cysts. The prevalence was between 3.7–4.9%.

The dimensional evolution of cysts is about one–two cm per year. Therefore, it often evolves asymptomatically to a considerable size. The symptoms depend on its location in different organs and the non-specific signs are anorexia, weight loss or asthenia. Possible therapies include surgical resection and anthelmintics medication. Secondary prophylaxis with anthelmintics such as Albendazole has a proven beneficial effect in the recurrence rate.

This case report follows the CARE Guidelines [5].

## 2. Patient Info and Clinical Findings

A 23-year-old woman was evaluated in the Emergency Unit with the following neurological symptoms: intense headache, intracranial hypertension syndrome (headache, vomiting), memory loss, emotional lability (frontal lobe signs), left eye blindness noticed a few hours before arriving in the Emergency Unit, with progressive visual impairments in the last three weeks and also a narrowed field of vision in the right eye. After the inspection of the cranial nerves and the ophthalmological examination, the clinical diagnosis was Foster Kennedy Syndrome. This sum of clinical findings can be associated with tumors in the frontal lobe, which result in optic nerve atrophy in the ipsilateral eye, disc edema in the contralateral eye, central scotoma (loss of vision in the middle of the visual fields) in the ipsilateral eye, anosmia (loss of smell) ipsilaterally [6], Figure 1.

The patient did not have any other pathological conditions. An important aspect of her evaluation was that she grew up and was living in a rural area, on a farm with many animals (sheep, dogs, cats) and was an active worker there.

## 3. Diagnostic Assessment and Diagnosis

A head computed tomography (CT) Scan with intravenous (IV) contrast (Figure 2 and Figure 3) showed a huge left frontal cyst, with a fine wall and a fine septum inside, minimum peripheral load, left frontal osteolysis, moderate peripheral edema, edematous optic nerves (bilateral secondary papillary edema). Cranio-cerebral magnetic resonance imaging (MRI) examination with contrast (Figure 4) supported the diagnostic suspicion of a cerebral hydatid cyst. Cystic lesions with membranes and daughter cysts are highly probable to be hydatid ones [7]. Based on the morphology classification, this hydatid cyst was a type IIa: a cyst with round daughter cysts at the periphery [8].

An abdominal ultrasound identified multiple hepatic cystic images, some septate, with dimensions from 30 to 60 mm. The imagistic investigation was continued with a Chest, Abdomen and Pelvic CT Scan with IV contrast, which led to identification of multiple hepatic cysts (more than 30) (Figure 5).

The hydatid cysts in the neck and head had a varied disposition, without an identified favorite area of implantation. However, only 2% of hydatid cysts were located in the skeleton, and only 3–4% of them were in the skull. Therefore, the incidence of an intraosseous localization at the level of the skull cap was below 0.08% [9].

For the present study, we will focus on the intracranial lesion, which belongs to our neurosurgical department. We will only make brief references to the treatment of liver cysts.

## 4. Therapeutic Interventions

Surgical intervention was performed urgently, without any delay, because of the acute and recent ophthalmological deficit. A bifrontal craniotomy was performed to evacuation three cystic formations of various sizes. The structure of the cysts was nonvascular, with a very friable fine wall, a clear aspect and a transparent liquid inside. The cysts were strictly extradural, with erosion of the internal plate of the left frontal bone. Unfortunately, the extravasation of the main cyst occurred at the time of the craniotomy (in the moment of lifting up the bone-flap (Figure 6). To lower the risk of cyst recurrence, abundant washing with H_2_O_2_ and saline solution was performed. Finally, we performed the replacement and fixation of the bone flap after bone scraping of the infiltrated area and a 20 min immersion in H_2_O_2_ and iodine (Figure 7 and Figure 8).

The postoperative evolution of the patient was a favorable one, with the transfer of the patient to the Abdominal Surgery department.

After the analysis of the abdominal surgery team, they decided to follow a drug treatment with Albendazole 400 mg twice a day for a 28-day cycle, followed by a liver cyst ablation surgery, during which 30 hepatic cysts were evacuated. Due to the countless milimetric cysts, the surgery could not have curative intentions; however, 60% of the liver parenchyma remained viable. For the moment, we can say that the goal was a palliative one, and the patient continued drug treatment with Albendazole 400 mg twice a day for another 28-day cycle, until the next follow-up.

The resected cystic membrane and its contents were sent for histopathological examination.

Hematoxylin and Eosin (HE) staining at 10× magnification and 20× magnification (Figure 9) showed the following: multiple histopathological membranous tissue fragments with the appearance of hyalinized, acellular structures, and inwards pale lamellar eosinophilic membranes with the appearance of proliferating membranes to which parasitic ovoid structures called scolices (the larval stage of the parasite) were attached. The surrounding fibro-conjunctival tissue contained a granulomatous-looking inflammatory infiltrate with the presence of multiple multinucleated giant cells and marked vascular congestion.

This histopathological examination, together with imagistic and biological examinations, established the definitive diagnosis, that of a hydatid cyst with intracranial localization [7].

## 5. Follow-Up and Outcomes

The first clinical and imaging follow-up at three months after the neurosurgical intervention and one month after the abdominal surgery found the patient with eating difficulties, anorexia, marked weight loss (approx. 12 kg), asthenia and fatigue. However, the neuro-ophthalmological examination showed an improvement of the visual acuity in both eyes, remitted headache syndrome, no intracranial hypertension syndrome and the net improvement of the patient’s cognitive performance [10].

The abdominal CT Scan identified important liver regeneration with a dimensional progression of 2 cysts. For these, it is expected that other surgeries will follow (Figure 10).

The CT scan of the head showed a normal postoperative aspect, without local recurrence of cystic lesions and with a good recovery of the left frontal parenchyma.

The 6 months and 1 year follow-up showed no recurrence of cystic masses and no neurological deficit (Figure 11).

## 6. Discussions

Hydatid cysts are the result of infection with the Tenia Echinococcus species and as a result cysts can form anywhere in the human body. However, only 0.5–2% of them are located in the skeleton and of these only 3–4% in the skull. Therefore, the case presented by us has an incidence of less than 0.08% among hydatid cysts and the treatments discussed in this article can be very helpful in preoperative planning [11].

Studying the literature, we noticed that the incidence of intracranial, extradural hydatid cysts is very low. We could only identify 12 cases, in which the localizations developed either directly from the calvarium or from the vessels of the extradural space, or by its inoculation from the intracerebral space through an apparently intact dura [12,13].

At the same time, only one case has been published so far of multiple hydatid cyst with extradural localization [14,15].

Of the cysts with intraosseous starting points and erosion of the internal plate towards the extradural space, only one case has been published with the intact removal of the cyst without its rupture. Thus, we may conclude that it is a real challenge to remove the cyst intact in cases of bone invasion [16,17].

In the Appendix A and Appendix B we have made an additional review table of the current publications on the topic of our article. This shows that there are very few studies on the subject. There are only seven articles referring to intracranial extradural or intraosseous hydatid cysts (some of them referring to several cases) published on the PubMed database. Although there are consistent publications referring to intracranial hydatid cysts in general, those related to intracranial or extradural cysts are few in number, obsolete and usually from underdeveloped countries. Many publications describe only radiological findings, with no details on the surgical procedure and follow-up. Also, even though the rate of rupture of extradural cysts during craniotomy is high, none of the articles address the possibility of the approach using a double concentric craniotomy [Appendix A and Appendix B] (Figure A1 and Figure A2).

Many publications describe extradural localizations in the spinal level, mainly because the bone volume of the spine is much larger than that of the skull, so the incidence is higher. Among the intraosseous intracranial localizations, we noticed that the higher incidence is at the level of the posterior fossa. An open question remains as to why the incidence is higher in the occipital bone.

Analyzing the series of cases presented in Duishanbai S et al. (2011), out of 97 intracranial hydatid cysts, only two had an extradural localization. This means an incidence of about 2% of all cranial hydatid cysts [Appendix A].

To summarize the therapeutic options that can be used in cased of hydatid cysts, we recommend the following: surgical excision, PAIR (puncture, aspiration, injection of protoscolicidal agent and respiration), chemotherapy with anthelmintic agent (Albendazole, Mebendazole), and conservative treatments, i.e., “watch and see”, which are useful in cases of inactive cysts. Small, heavily calcified cysts can be considered inactive or dead and can only be monitored periodically. This must take into account the location of the cyst and the risk of rupture caused by an internal or external force [18].

With a very high absorption and bioavailability, Albendazole is recommended by the WHO at a dose of 10–15 mg/kg/day for at least three months [19,20]. All these therapeutic methods can be applied to hydatid cysts regardless of location, including those at the intracranial level.

Among surgical techniques, we recommend the Dowling technique, in which after performing the craniotomy, a corticectomy of ¾ is performed above the cyst followed by the instillation of saline solution between the cyst and the brain. This technique is the most common and has a good result.

In terms of craniotomy approach, according to our experience, we propose that in cases of hydatid cysts with the invasion of the internal bone plate, a double concentric craniotomy should be performed. It is a very good and useful way to preserve the cysts intact during the craniotomy. In such cases, this is the most important surgical moment. This technique is described and used in intracerebral tumors with bone invasion [21].

A variant of the PAIR technique combined with anthelmintic treatment could be applied in neurosurgery using endoscopic treatment. There are still no reported cases in the literature of intracerebral hydatid cysts excised by endoscopic treatment, but it may be an option to consider.

In the diagnostic process of the hydatid cyst, eosinophilia has only a limited value. However, eosinophilia has an important prognostic value because it is observed only in cases of ruptured or leaking cysts [22].

In the case of our patient, the abdominal surgeon’s choice in the timing of the liver surgery after offering the viable liver tissue a period of time to regenerate under treatment with Albendazole is highly admirable and masterly. At the time of the first evaluation, the viable liver parenchyma was very limited and divided.

## 7. Conclusions

Although the incidence of hydatid cysts in the European Union and the United States is not very high, with increased migration and international transit, we can expect them to appear anywhere in the world. Even if sometimes the surgery cannot be postponed, there must be a wide discussion based on the surgical approach to such cases and the best therapeutic options. The choice of the surgeon can make the difference between a perfect operation and one with catastrophic results.

## Figures and Tables

**Figure 1 brainsci-11-01604-f001:**
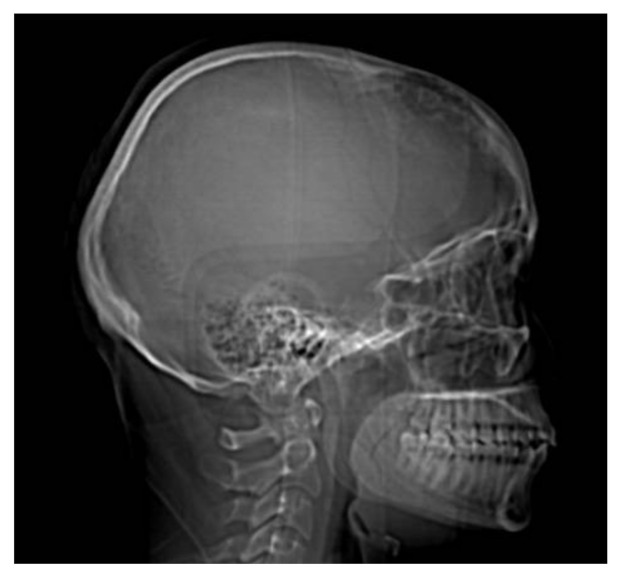
Skull radiography—lateral view. Erosion of the inner table of the frontal bone.

**Figure 2 brainsci-11-01604-f002:**
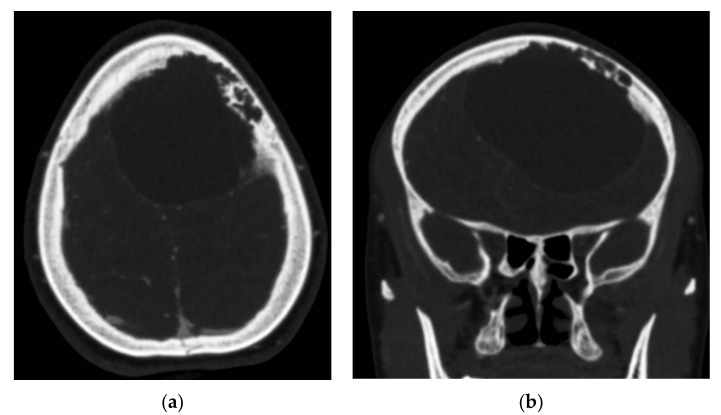
Head CT Scan bone window. Erosion of the frontal bone inner table. (**a**) axial; (**b**) coronal.

**Figure 3 brainsci-11-01604-f003:**
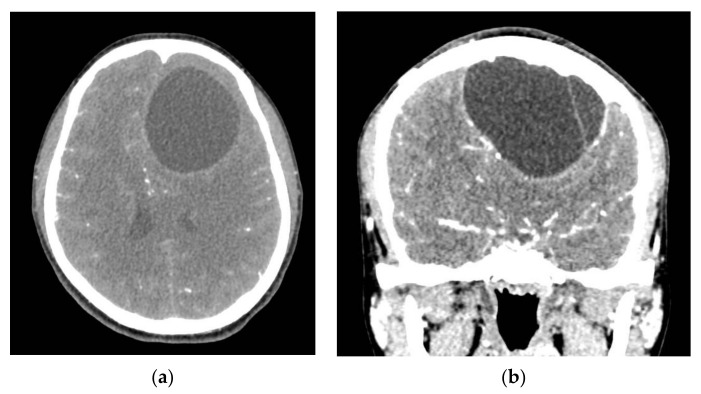
Cranial computed tomography (CT) scans with intravenous (IV) contrast. (**a**) axial; (**b**) coronal.

**Figure 4 brainsci-11-01604-f004:**
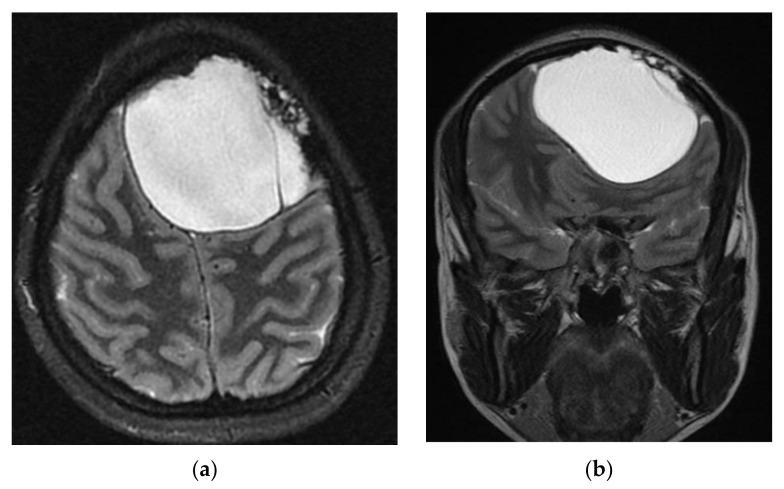
Cerebral magnetic resonance imaging (MRI) supports the diagnostic suspicion of intracerebral hydatid cyst: (**a**) axial; (**b**) coronal.

**Figure 5 brainsci-11-01604-f005:**
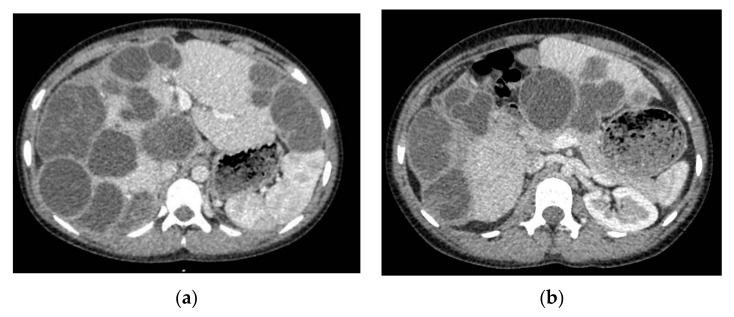
Abdominal CT Scan with IV contrast. (**a**,**b**) multiple liver cystic lesions with small areas of normal parenchyma.

**Figure 6 brainsci-11-01604-f006:**
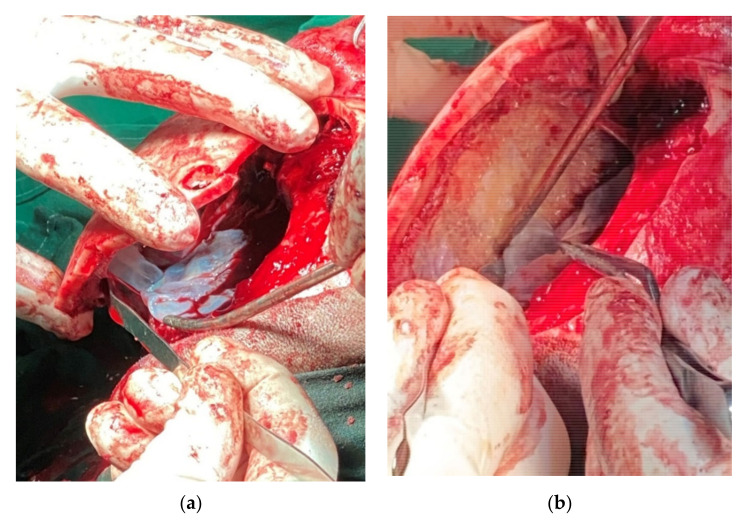
Intraoperative view. Cyst extravasation at the moment of craniotomy. (**a**) cyst rupture at the time of the craniotomy; (**b**) the cyst membrane infiltrates the frontal bone.

**Figure 7 brainsci-11-01604-f007:**
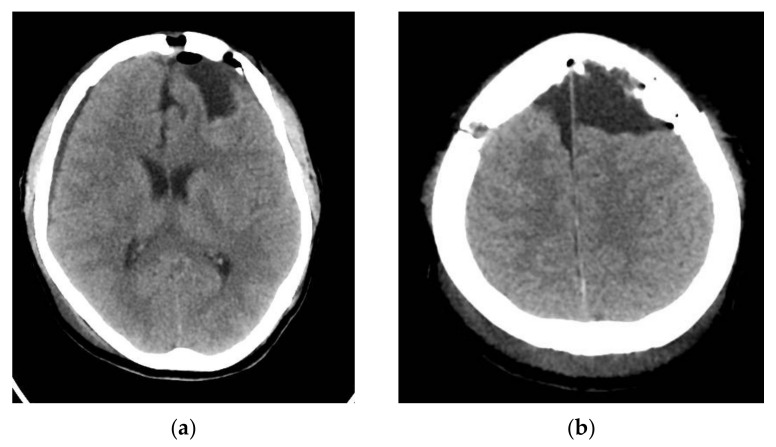
Native head CT Scan 24 h after surgery. (**a**,**b**) axial slices.

**Figure 8 brainsci-11-01604-f008:**
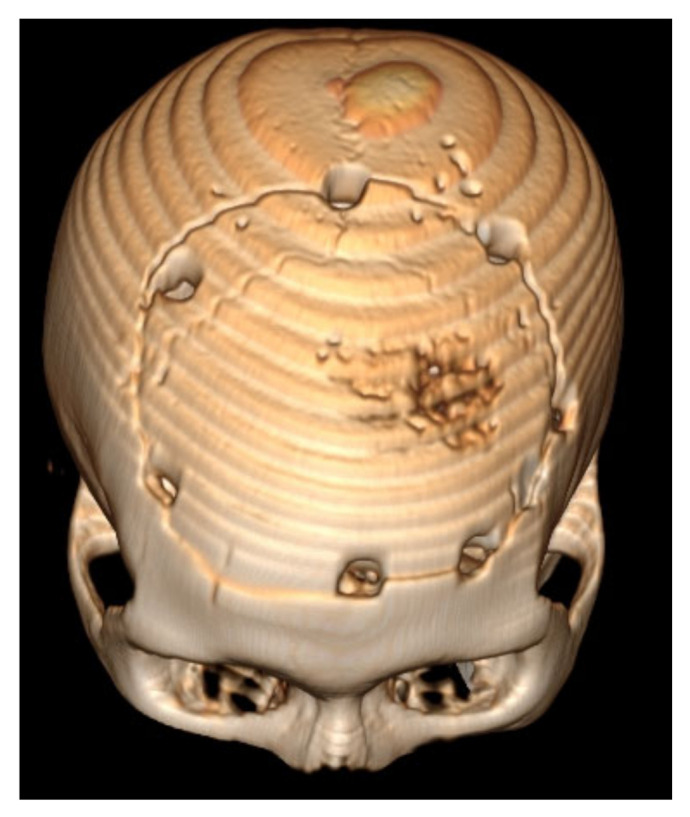
3D recon of the skull.

**Figure 9 brainsci-11-01604-f009:**
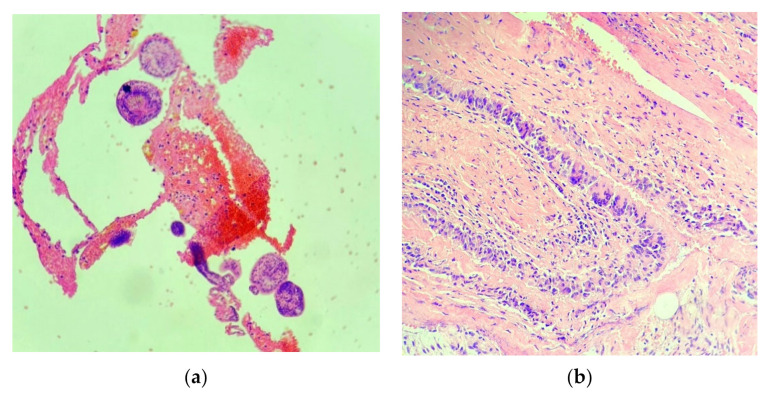
Hematoxylin and Eosin (HE) staining: (**a**) 10× magnification and (**b**) 20× magnification.

**Figure 10 brainsci-11-01604-f010:**
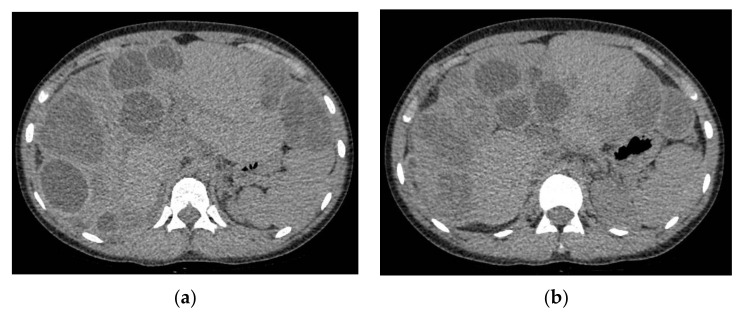
Abdominal CT scan. (**a**,**b**) we can see the areas with viable parenchyma improved compared with the previous abdominal CT Scan.

**Figure 11 brainsci-11-01604-f011:**
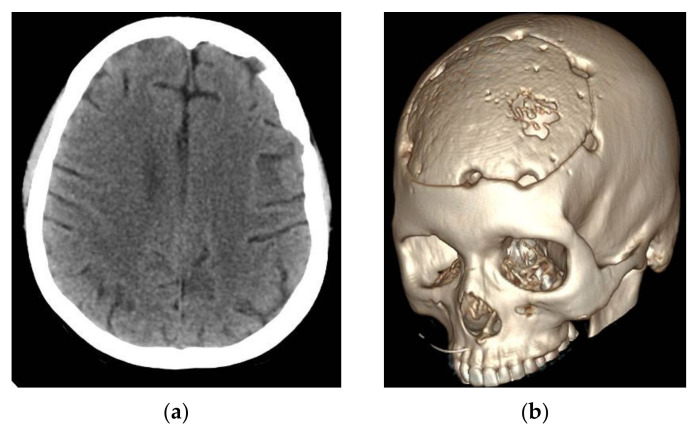
Twelve months follow-up CT Scan. (**a**) axial—very good left frontal lobe recovery. (**b**) 3D reconstruction.

## Data Availability

Not applicable.

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
