# Peer review of "Intraoperative Rupture of an Intracranial, Extradural Hydatid Cyst: Case Report and Treatment Options"

_brainsci, 2021, doi:10.3390/brainsci11121604_

Round 1

Reviewer 1 Report

In the reviewed article, the authors describe the case of a patient treated surgically due to an intracranial epidural cyst, which was diagnosed as echinococcosis. The diagnosis was based on imaging tests, i.e. cerebral CT and MRI as well as ultrasound and abdominal CT scans. Despite careful reading, I found no other study to support the diagnosis. The lack of histopatologic, serologic, and immunologic studies significantly reduces the value of the work.   The CT and MR appearance is not always diagnostic, and differentiation from other epidural cystic lesions may not always be possible.   "The definite diagnosis of hydatid cyst must depend on pathological examination of the surgical specimen." see 7. Limaiem, F., Bellil, S., Bellil, K., Chelly, I., Mekni, A., Kallel, J., Haouet, S., Zitouna, M., & Kchir, N. (2009 ). Hydatid cyst of the 233 cranial vault. J Infect Dev Ctries, 3 (10), 807-810. https://doi.org/10.3855/jidc.48   The surgical technique and pharmacological treatment described by the authors are typical can be readily found in other publications. The lack of originality is the second drawback of this work.

Reviewer 2 Report

Interesting case. Out of curiosity, was the cyst wall/content sent to the pathology lab for histology or even cultures?It would be good to include that in the report.

Reviewer 3 Report

This case study reported the clinical course of a patient with extremely rare abdominal hydatid cysts and cerebral/skull extradural bone invasion. The authors treated the brain condition satisfactorily with an acceptable duration of follow-up. The language applied throughout the manuscript is coherent. I have the following comments for the authors to improve the manuscript.

  • It would be more persuasive to add a series of illustrative histopathological photographs to demonstrate the pathognomonic features of infection process of this specific parasite.

  • As stated by the authors, some previous papers have already been reported regarding the condition. I would recommend the authors to add an additional review table including all reported cases that discuss on the same problem with citing references to support that this is an unique rare case worth to be presented and published to raise the awareness of the clinician. It may then undoubtedly add contribution to the scarce literature on this particular topic.

Round 2

Reviewer 1 Report

After completion, the article looks much better. The added histopathological examination was necessary to confirm the diagnosis. The added appendix also increases the value of the work and makes it more interesting for the reader.
Information on the immunity status of the patient would be a significant supplement to the clinical data in the context of the case.
It would also be useful to briefly discuss the importance of serological testing in diagnosing disease.